# Moringa Leaf Extract Mitigates the Adverse Impacts of Drought and Improves the Yield and Grain Quality of Rice through Enhanced Physiological, Biochemical, and Antioxidant Activities

**DOI:** 10.3390/plants12132511

**Published:** 2023-06-30

**Authors:** Shahbaz Khan, Danish Ibrar, Zuhair Hasnain, Muhammad Nawaz, Afroz Rais, Sami Ullah, Safia Gul, Manzer H. Siddiqui, Sohail Irshad

**Affiliations:** 1Department of Agronomy, Ghazi University, Dera Ghazi Khan 32200, Pakistan; 2Colorado Water Center, Colorado State University, Fort Collins 80523, CO, USA; 3Plant Genetic Resources Institute, National Agricultural Research Centre, Islamabad 45500, Pakistan; 4Department of Agronomy, Pir Mehr Ali Shah Arid Agriculture University, Rawalpindi 46000, Pakistan; 5Department of Agricultural Engineering, Khwaja Fareed University of Engineering and Information Technology, Rahim Yar Khan 64200, Pakistan; dmnawaz@kfueit.edu.pk; 6Department of Botany, Sardar Bahadur Khan Women’s University, Quetta 1800, Pakistan; rais_botany@yahoo.com (A.R.); sgul53@yahoo.com (S.G.); 7Pakistan Agricultural Research Council, Arid Zone Research Centre, Dera Ismail Khan 29120, Pakistan; samiullahms89@yahoo.com; 8Department of Botany and Microbiology, College of Science, King Saud University, Riyadh 11451, Saudi Arabia; mhsiddiqui@ksu.edu.sa; 9Department of Agronomy, MNS-University of Agriculture, Multan 64200, Pakistan; sohailuaf99@gmail.com

**Keywords:** biostimulant, foliar application, growth, productivity, water deficit

## Abstract

Agriculture, around the globe, is facing great challenges including the need to increase the production of nutrient-dense food and to withstand climate change’s impact on water and soil conservation. Among these challenges, drought stress is considered the most overwhelming danger for the agriculture sector. Organic plant growth ingredients are frequently used to enhance the growth and production of field crops cultivated in normal and unfavorable conditions. The present study was designed to explore whether leaves extracted from various landraces of Moringa could play a defensive role against drought stress in rice. Seedlings were grown under three water conditions, i.e., normal conditions (control; 100% field capacity), moderate (75%), and severe drought (50%). Leaf extracts obtained from four Moringa landraces were used as foliar spray at the tillering, panicle initiation, and grain filling stages of cultivating rice plants. The levels of water stress negatively influenced photosynthetic pigment synthesis, gas exchange traits, antioxidant activities, and yield and grain quality parameters. Leaf extracts, at the rate of 3%, from all the landraces significantly enhanced the biochemical, physiological, and yield-related attributes of rice plants under normal and unfavorable growth conditions. Particularly, leaf extract from the Faisalabad landrace was the most effective biostimulant to increase photosynthetic (8.2%) and transpiration (13.3%) rates, stomatal conductance (8.3%), chlorophyll *a* (15.9%) and *b* (9.7%) contents, and carotenoids (10.4%) as compared to water spray. The maximum photosynthesis rate was observed at 14.27 µmol CO_2_ m^−2^ s^−1^ via application of leaf extract from the Faisalabad landrace followed by the DG Khan (13.92 µmol CO_2_ m^−2^ s^−1^) and Multan (13.9 µmol CO_2_ m^−2^ s^−1^) landraces, respectively. Improved grain yield (25.4%) and grain quality (an increase of 10.1% in amylose with a decrease of 2.8% in amylopectin) in rice plants along with enzymatic activities such as catalase (21.2%), superoxide dismutase (38.6%), and ascorbate peroxidase (24.3%) were observed at the peak after application of leaf extract from the Faisalabad landrace. The maximum grain yield of 53.59 g per plant was recorded when using Faisalabad landrace leaf extract and the minimum (40 g) using water spray. It is concluded from the findings of the current experiment that leaf extract from the Faisalabad landrace possesses higher biostimulant potential than other landraces and can be applied to mitigate the adverse impacts of drought stress with higher productivity and improved grain quality of rice.

## 1. Introduction

Variability in climatic patterns and the unavailability of water and food along with unprecedented population growth are a few of the emerging challenges being faced by the farming community. Among them, water scarcity is the most dangerous in terms of adversely affecting crop productivity, particularly in the dryland agricultural areas of Asia and the Pacific, which cover more than 223 million hectares of land [1]. Unavailability of proper moisture for plants induces a reduction in photosynthetic pigments, which in turn reduces photosynthesis activity [2]. Further devastating drought effects on plants include reduced CO_2_ uptake, impaired cell elongation and division, and an imbalance between antioxidants and reactive oxygen species [3]. Rice is the staple food of more than 3 billion people worldwide and provides 20% of the world’s energy supply; wheat and maize provide 19% and 5%, respectively [4]. Worldwide, rice grain yield was recorded as 502.98 million tons from an area of 165.25 million hectares in 2022 [5]. Water shortage and seedling stress severely affect the roots of the rice plant, as stress at this stage reduces cell expansion and other properties of the root development system [6]. Drought stress causes the production of reactive oxygen species (ROS) such as hydrogen peroxide and superoxide which damage the carbohydrates, protein, lipids, DNA, and nucleic acids in rice plants [3,7]. However, to detoxify ROS, various enzymatic antioxidants such as catalase (CAT), super dismutase (SOD), and peroxidase (POD) along with non-enzymatic antioxidants play defensive roles, and the ROS are degraded. Drought stress is very damaging during grain propagation and grain filling as it hampers pollination through damaging pollen viability thus reducing grain formation. Drought stress also reduces the rate and time of grain filling, which is ultimately responsible for a reduction in grain yield [8].

Plants are equipped by nature with various mechanisms and strategies to mitigate the effects of drought stress, e.g., through increasing the concentration of compatible solutes, reducing the stomatal activities, and through the activation of defensive mechanisms [9]. In addition to the instinctive abilities of plants, different crop husbandry measures can be adopted to further lower the negative impact of drought stress on plants. These include the cultivation of drought tolerant/resistant varieties, the use of mulches (natural/synthetic) [10,11], stress signaling molecules, osmoprotectants, and extracts from crops/plants [12,13,14]. In recent years, it has been shown that extracts from crops/plants are biologically safe and economically viable and have great potential for improving crops in stress environments [12]. Aqueous extracts of various crop plant parts have been identified to improve plant performance in normal and unfavorable circumstances through modifying the metabolism of phytohormones, photosynthetic activities, the uptake of nutrients and water, signal transduction, the growth and development of gene expression, leaf aging, and grain distribution [15]. An aqueous extract of the annual broad-leaved herb *Achyranthes aspera* L. is also used in agriculture due to its allelopathic potential [16]. The use of biostimulants, especially humic acid along with boron and zinc, was found to be responsive regarding improved growth and yield attributes [17].

Moringa (*Moringa oliefera* L.) is of great interest to plant scientists and biologists, as its leaves are a rich source of minerals, antioxidants, vitamins, and growth-promoting hormones [12,18]. Moringa leaves are an excellent allelopathic culture, and their extract has been applied because of several allelochemicals present in their culture [19,20]. *Moringa oleifera* leaf extract (MLE) application increases seedling establishment, plant growth, and productivity in crop plants under both stress and normal conditions [21,22], and it can also improve the quality of produce [23]. This study was designed to investigate the potential benefits of different Moringa oleifera species (exotic and local) in improving the growth, yield, quality, and biochemical properties of rice grains grown under normal and water-stressed conditions. The outcomes of this study will assist in rice yield and productivity under limited water/drought stress conditions.

## 2. Results

Table 1 shows that all the parameters of rice plants, which include gaseous exchange, enzymatic activities, grain production, grain quality, and physiological attributes, were at significant levels in both factors, i.e., drought stress and MLE application. Figure 1 depicted the effects of MLE application on photosynthesis (*A*), respiration (*E*), and stomatal conductance to water (*gs*) of local and exotic breeds under both normal and water-deficient conditions. Physiological parameters such as photosynthetic rate, transpiration rate, and stomatal conductance were significantly different for various drought treatments.

The effect of MLE treatments on gas exchange attributes of rice cultivated under different water regimes is shown in Figure 1. The photosynthetic rate, stomatal conductance, and respiration rate were observed at the peak in case of CC followed by MDS. On the other hand, SDS adversely influenced the earlier mentioned attributes. The maximum biostimulant potential was observed in those plants which were treated with MLE-LF regarding the photosynthetic rate and stomatal conductance, followed by MLE-LD and MLE-LM (Figure 1a,b). In the case of respiration rate, MLE-LF was also found to be the most effective among all the MLE treatments (Figure 1c).

The effects of foliar treatments with MLE regarding chlorophyll contents and carotenoids are shown in Figure 2. Significant differences were observed in chlorophyll contents and carotenoids in response to drought stress treatments. In addition, MLE of local landraces was more effective than exotic landraces regarding chlorophyll contents and carotenoids. The interaction of both factors, MLE and drought, was also observed as statistically significant. Maximum improvement in chlorophyll contents and carotenoids was regarded after the application of MLE-LF under CC while maximum reduction was found under severe stress conditions with water spray (Figure 2a–d). MLE treatments were also observed to be effective under MDS and SDS.

Figure 3 reveals the impact of *Moringa oleifera* leaf extract on the enzymatic action and H_2_O_2_ under various degrees of drought stress. In the case of extreme drought stress with MLE-LF, the SOD activity was most remarkable and it was statistically comparable to extracts of local and exotic cultivars. In contrast, the lowest SOD activity was recorded under water spray control circumstances. There was a considerable divergence in the activity of catalase and ascorbate peroxide in response to the different levels of drought stress among the foliar treatments. The catalase and ascorbate peroxide activity was highest in MLE-LF under extreme drought stress and least in water spray with extreme drought stress. Severe drought stress with water splashes had the maximum amount of H_2_O_2_ content, while the other MLE treatments under control conditions produced the lowest amount of H_2_O_2_ (Figure 3a–d).

The data for the height of the plants is depicted in Figure 4. The highest increment in height was observed in MLE-LF, followed by MLE-LD and MLE-LM. The lowest plant height was recorded in WS. The tallest plants were seen in CC followed by MDS, while the shortest plants were observed in SDS. Maximum productive tillers were observed in CC followed by SDS. MLE from all the local landraces increased the number of productive tillers per plant when compared to WS; however, MLE-LF had the highest number of productive tillers per plant, followed by MLE-LD, MLE-LM, and MLE-EL, respectively (Figure 4). Notable differences in panicle length were recorded under various levels of drought stress. The maximum panicle length was observed in CC followed by MDS, whereas the minimum panicle length was observed in SDS. MLE-LF notably increased the length of rice panicle, whereas the lowest length was measured in WS (Figure 4). The panicle showed a considerable disparity in the number of grains in various drought conditions. The CC presented the highest number of grains per panicle, followed by MDS, and the least by SDS (Figure 5). The MLE-LF notably increased the number of grains per panicle, whereas WS had the lowest count of the aforesaid attribute (Figure 4). The highest rice yield was recorded with CC followed by MDS, and the lowest with SDS. From the different local landraces of MLE, MLE-LF had the highest grain yield, followed by MLE-LM, MLE-LD, and MLE-EL, respectively (Figure 5).

Among rice grains, CC had the highest amylose content (%), followed by MDS, while SDS had the lowest amylose content (Figure 6). The MLE-LF had the greatest amylose content among the MLEs of different local landraces, followed by MLE-LM, MLE-LD, and MLE-EL, respectively (Figure 6). Drought stress and MLE treatments greatly impacted the amylopectin content (%). The amylopectin content (%) was highest in WS and lowest in MLE-LF, respectively. Similarly, it was observed that the MLE from exotic breeds had a lower amylopectin concentration (%) than the WS (Figure 6).

## 3. Discussion

The results of the present study confirmed the use of Moringa leaf extract (MLE) significantly increased the gaseous exchange characteristics, physiological responses, enzyme activity, and productivity of rice crops under drought stress. Results on gaseous exchange properties were consistent with previous studies which found that photosynthesis rate (*A*), stomatal conductance (*gs*), and respiration rate were significantly reduced due to drought stress [24,25]. The foliar application of MLE was observed to be beneficial for improving the rice plant’s performance under water deficit conditions. MLE is a rich source of mineral elements, secondary metabolites, and growth ohormones, particularly cytokinin; zeatin [18,26] which may stimulate the endogenous levels of compatible solutes and secondary metabolites and improve the plant growth through maintaining the water relations and partitioning of photosynthates. Although *A* and other related activities in flag leaves are important sources for enhancement in crop production, these are also susceptible to environment-induced stress, including drought stress [27]. Photosynthesis rate (*A*), stomatal conductance (*gs*), and respiration rate (*E*) were also significantly reduced under water deficit conditions during the current study (Figure 1a–c). The reduction in gas exchange attributes is linked with the availability of water contents; as the water level drops, the plant may regulate the defensive mechanism to overcome the water loss through closing its stomata. Liu et al. [28] also observed that drought stress reduced the net *A* and *gs* of rice. Closing the leaf stomata reduces CO_2_ uptake, thus reducing the size of the source. In addition, leaf senescence may also be a limiting factor in the conversion of dry matter into panicles during the milking phase [29,30].

Disturbed physiological activity may be responsible for low grain yields. Drought stress caused a reduction in chlorophyll pigments and carotenoids (Figure 2a-d). As the intensity of drought stress increases, concentrations of these dyes decreases linearly. In this study, photosynthetic pigments showed a reduction under drought stress. These results are in line with Swapna et al. [31], who indicated that the chlorophyll along with carotenoid content of rice plants decreased during drought stress. It is also reported that the imposition of drought stress results in the reduction of chlorophyll and other photosynthetic pigments in rice plants [2]. There are also reports that drought can damage the chlorophyll content. The content of chlorophyll was higher in the non-stressed environment than in the drought-stressed environment [32]. It was also found that drought stress significantly reduces chlorophyll content, and chlorophyll content decreases with increasing drought intensity [33]. Irshad et al. [34] reported that the application of MLE increased the chlorophyll contents in Kabuli chickpeas. The application of MLE is also responsible for enhancing the concentration and activity of photosynthetic pigments in wheat and quinoa [12,35]. Chlorophyll and other photosynthesis-related parameters can be used to assess the severity of drought stress being faced by the rice plant [36]. Drought also induced a reduction in the rate of photosynthesis, H_2_O content, and transpiration rate in conjunction with increased stomatal resistance [37].

Foliar-applied MLE on rice plants grown under drought stress significantly enhances their ability for gaseous exchange and the concentration of photosynthetic pigments (Figure 1 and Figure 2). This positive effect of MLE may be attributed to the significant concentration of specific phytochromes with proven antioxidant properties, such as chlorophyll and carotenoids (xanthine, beta-carotene, alpha-carotene, and lutein) in leaves of Moringa plant [38]. In addition, Moringa leaves contain many macronutrients that increase the chlorophyll (*a* and *b*) concentration in rice. MLE has also been reported to stimulate early cytokinin formation; therefore, premature aging of the leaves is avoided, resulting in an increase in leaf surface area and an increase in the number of photosynthetic pigments [39]. Our results are also in concurrence with Khan et al. [40], who reported a significant improvement in photosynthetic pigments of wheat following the application of MLE under favorable growing conditions. Foliar-applied MLE during tillering and topping phases enhances chlorophyll *a* and *b* concentrations in wheat plants [41]. Enhancement in growth due to foliar application of MLE on plants may be accredited to the presence of various metabolites and allelochemicals such as phenolic compounds, ascorbate, and zeatin [18]. Khan et al. [40] also reported the same findings as observed in the present study, wherein they reported that the *Moringa oleifera* leaf extract (MLE) of Faisalabad origin (MLE-LF) showed higher bio-stimulating potential, probably due to the presence of higher concentrations of bio-stimulating elements, substances promoting plant growth, minerals, and antioxidants.

A plant’s ability to mitigate stressful environments is based upon antioxidant-related defense activities. Application of leaf extracts derived from Moringa oleifera induced enzymatic antioxidants, especially under drought conditions. The activity of enzymatic antioxidants showed a linear relationship with the onset of drought stress (Figure 3a–d). Our findings suggest that the increase in superoxide dismutase, catalase, and ascorbate peroxidase activity may be related to the initiation of antioxidant reactions that provide a defensive shield to plants against oxidative damage. According to Foyer and Noctor [42], plants use the induction of enzymatic antioxidant activity as a natural adaptation against oxidative stress. Superoxide dismutase has been adopted as the basis of defense in responses to oxidative damage [43]. Overexpression of superoxide dismutase is considered an important drought resistance mechanism when accompanied by an enhanced H_2_O_2_ capture mechanism [44]. Hanafy [45] reported a considerable boost in the activity of enzymatic antioxidants (GR, SOD, and APX) in soybeans subjected to drought stress. The induction of superoxide dismutase and ascorbate peroxidase-related activities in oilseed rape (*Brassica napus* L.) are also increased in water-scarce environment [46]. The use of MLE caused a significant increase in superoxide dismutase activity followed by glutathione reductase and ascorbate peroxidase in soybeans, respectively. Zaki and Rady [47] also observed that the use of MLE in the form of seed soaking or foliar spraying increased the activity of enzymatic antioxidants such as glutathione reductase, superoxide dismutase, and ascorbate peroxidase in bean plants (*Phaseolus vulgaris* L.).

This study describes the effects of the growing environment, especially drought stress, on the growth, productivity, yield, and grain quality of rice crops. An increased level of drought stress had a direct impact on yield and other yield-contributing characteristics (Figure 4 and Figure 5). This decrease in rice plant growth and production capacity is related to the reduction in the photosynthetic capacity of the leaves and photosynthetic surfaces under conditions of limited water availability [48]. Several drought conditions adversely affect enzyme activity and thus reduce the growth and yield of tomato crops [49]. These findings are in line with Mukamuhirwa et al. [50], who suggested that drought stress significantly reduces the growth parameters of rice plants because the unavailability of nutrient and water adversely affect biochemical processes and the synthesis of structural compounds. Farooq et al. [51] observed that water scarcity limits plant growth through affecting many indispensable biochemical and physiological processes and pathways such as respiration, photosynthesis, ion uptake, translation, nutrient assimilation, and carbohydrate metabolism. Water is considered a universal solvent and essential for most of the biochemical processes occurring in plants simultaneously. Exogenous application of MLE from all land breeds substantially improved growth parameters and yields even under drought stress. However, under conditions of abundant water and/or drought stress, foliar spraying of MLE from Faisalabad cultivars resulted in the greatest increase in grain yield (Figure 5). Barnabas et al. [52] argued that drought stress significantly reduced grain weight due to a reduced number of endosperm cells, reduced starch synthesis, and reduced transport of photosynthetic products from source to mouth [53]. Farooq et al. [54] also studied the improvement of root and shoot length and biomass of wheat and chickpeas via exogenous application of aqueous brassica extract.

Foliar-applied MLE considerably improved the performance of rice plants under water-scarce conditions, as evidenced by the improvement in grain number, grain weight, and yield. These results are in concurrence with those reported by earlier studies by Basra et al. [55], Yasmeen et al. [56], and Khan et al. [20,40], who reported that the presence of ascorbic acid, phenolics, zeatin, carotenoids, antioxidants, essential phyto components, and vitamins in MLE is responsible for improved growth and yield of field crops. Studies have shown that the use of MLE as an exogenous foliar spray can increase the level of endogenous hormones and thus promote plant growth [20,57]. Cytokines have been found to play an important role in promoting chlorophyll biosynthesis, cell division, and cell elongation in MLE [58]. Drought stress negatively affected the content of amylose, amylopectin, and protein in wheat, while MLE improved these parameters (Figure 6). These results are consistent with the results of Mukamuhirwa et al. [59], who indicated that drought had a negative effect on amylose content in cereals. The amylose content ranged from 14% to 25%, which corresponds to intermediate amylose content [60].

## 4. Materials and Methods

### 4.1. Experimental Specifics

The current experiment was designed to compare the biostimulant potential of leaves extract obtained from four Moringa landraces on rice plants. Four landraces—Multan landrace, Faisalabad landraces, Dera Ghazi Khan landrace, and Indian landrace—of *Moringa oleifera* were selected for current experimentation. The basmati Pak cultivar of rice was used as a test crop and its seeds were purchased from a well-known company in Punjab Province, Punjab Seed Corporation. In 2019, the experiment was carried out in the greenhouse of the Faculty of Agricultural Sciences, Ghazi University, Dera Ghazi Khan 32200, Pakistan (30.05°N, 70.64°E, 129 m above sea level). Clay pots (height 45 cm, diameter 30 cm), having growth medium, were used for the cultivation of rice seedlings. The growth medium consists of equal parts of compost, sand, silt, and clay with an EC and pH of 2.59 dS m^−1^ and 7.7, respectively. Ten seeds were sown in each plastic pot followed by thinning after 15 days of seedling emergence to maintain four seedlings per pot for further experimentation.

The current experiment was conducted according to a completely randomized design (CRD) with the factorial arrangement primarily due to two factors: 1) drought levers and 2) MLE foliar treatments. There were three levels of drought stress, i.e., control conditions (CC), moderate drought stress (MDS), and severe drought stress (SDS). Moringa leaf extract (MLE) application consisted of five treatments: i) water spray (WS), ii) MLE from Multan landrace (MLE-LM), iii) MLE from Faisalabad landrace (MLE-LF), iv) MLE from Dera Ghazi Khan Landrace (MLE-LD), and v) MLE from exotic landrace of India (MLE EL).

### 4.2. Drought Stress Imposition

There were three levels of drought, i.e., control conditions (CC), moderate drought stress (MDS), and severe drought stress (SDS). Drought treatments: CC, MDS, and SDS, were induced through limiting the water supply to keep the field capacity of pots at 100%, 75%, and 50%, respectively. Field capacity was determined via the gravimetric method according to Nachabe [61]; pots were weighed regularly daily and a measured amount of water was added accordingly. Furthermore, rice plants were grown in their respective treatments throughout experimentation from transplanting to maturity.

### 4.3. Extract Preparation and Application

Mature, fresh, healthy, and disease-free leaves from the trees of respective *Moringa oleifera* landraces: Multan landrace (Latitude: 30°9′54.79″ N, Longitude: 71°29′48.72″ E), Faisalabad landrace (Latitude: 31.43°26′5″ N, Longitude: 73.06°4′7″ E), Dera Ghazi Khan Landrace (Latitude: 30°03′22.10″ N, Longitude: 70°38′5.17″ E) and an exotic landrace of India (Latitude: 31.43°26′5″ N, Longitude: 73.06°4′7″ E). The exotic landrace is well established at the research farm area of the Agronomy Department, University of Agriculture, Faisalabad, Pakistan. Before overnight freezing, collected leaves were rinsed with tap water to move the dust. After storage overnight in a refrigerator, extraction was done with the help of a locally assembled machine according to already established protocol with little modification [20]. The extract was sieved with distilled water to obtain a solution of 3% concentration. Moringa leaf extract (MLE) treatments included water spray (WS), MLE from Multan landrace (MLE-LM), MLE from Faisalabad landrace (MLE-LF), MLE from Dera Ghazi Khan Landrace (MLE-LD) and MLE from exotic landrace of India (MLE-EL). The MLEs in all the treatments were applied thrice during experimentation at tillering, panicle initiation, and grain-filling stages of rice plants. A hand sprayer of one-liter capacity was used to apply the foliar spray. WS treatment was considered as a control treatment to compare with other MLE treatments.

### 4.4. Determination of Physiological Parameters

Samples were collected from the flag leaf of rice plants after one week of foliar application of treatments at the grain-filling stage. The procedure of Arnon [62] was adopted to determine the chlorophyll *a*, *b,* and carotenoid contents at the absorbance of 663, 645, and 480 nm using the spectrophotometer. The following formulas were under consideration:Chlorophyll *a* = [12.7 (OD at 663) − 2.69 (OD at 645)] × V/1000 × W
Chlorophyll *b* = [12.7 (OD at 645) − 4.68 (OD at 663)] × V/1000 × W
Carotenoids = [(OD at 480) + 0.114 (OD at 663) − 0.638 (OD at 645)] × V/1000 × W
where OD = optical density, V = volume of extract (ml), and W = weight of fresh leaves (g).

The sum of chlorophyll *a* and *b* was considered to record the total chlorophyll contents.

### 4.5. Measurement of Gas Exchange Characteristics

Data regarding the gas exchange characteristics were collected after one weak foliar application of treatments at the grain-filling stage of rice plants. A portable gas analyzer (LI-6400) was used according to Long et al. [63] from 10:00 to 12:00 of the day to measure the photosynthesis rate (*A*; µmol CO_2_ m^−2^ s^−1^), respiration rate (*E*: mmol H_2_O m^−2^ s^−1^) and stomatal conductance (*gs*; mmol m^−2^ s^−1^).

### 4.6. Estimation of Enzymatic Attributes

Similarly, samples were collected from the flag leaf of rice plants after one weak foliar application of treatments at the grain-filling stage. Spectrophotometer was used to observe the enzymatic activities. The activity of superoxide dismutase (SOD) was checked using the protocol of Giannopolitis and Ries [64]. In cuvettes, 50 µL enzyme extract, 50 µL NBT (nitroblue tetrazolium), 50 µL riboflavin, 100 µL methionine, 250 µL phosphate buffer, 40 µL distilled water, and 100 µL Triton X was added and kept it under light for 15 min. After that, the absorbance was noted at 560 nm using a spectrophotometer. Catalase activity (CAT) was measured following the method of Chance and Maehly [65]. In cuvette 100 µL enzyme extract, 1.9 mL phosphate buffer, and 1 mL H2O2 (5.9 mM) were added. Using a spectrophotometer, absorbance was taken at 240 nm. The activity of ascorbate peroxidase (APX) was estimated with slight modifications by Nakano and Asada [66] using a reaction mixture (1 mL) containing 50 mM potassium phosphate (pH 7), 0.1 mM hydrogen peroxide and 0.5 mM ascorbate. The hydrogen peroxide was measured following the protocol of Velikova et al. [67]. Fresh leaves (0.1 g) were grinded in 5 mL of 0.1% (w/v) trichloroacetic acid (TCA), in a pre-chilled pestle and mortar. Then, homogenized leaf samples were centrifuged for 15 min. After that 0.5 mL supernatant was thoroughly mixed with 0.5 mL potassium phosphate buffer (pH 7) and 1 mL of potassium iodide solution. The absorbance was noted at 390 nm using a spectrophotometer.

### 4.7. Measurement of Agronomic and Yield Characteristics

The height of rice plants was recorded at maturity with the help of a meter rod. The total number of productive tillers per seedling was manually counted from each experimental unit. The ear length of five randomly selected panicles was recorded using a scale. For grain per ear, five ears were taken from each treatment and the number of grains was counted manually and the average was calculated. The thousand-grain weight was obtained through weighing 1000 grains with an electric balance. To determine the grain yield, fully mature plants were harvested from each pot and grains were collected through threshing the plant. Grains were weighed using an electric balance and grain yield was recorded.

### 4.8. Assessment of Quality Traits

The content of amylopectin and amylose in rice was determined according to the protocol proposed by Zhu [68]. Latimer [69] proposed methods was adopted to estimate the protein contents using the Microkjeldahl apparatus.

### 4.9. Statistical Analysis

A statistical package “Statistics 8.1” was used to analyze the collected data of various parameters. Microsoft Excel was used to calculate the means and standard errors for the graphical presentation. ANOVA technique (Fisher’s analysis of variance) was applied to demonstrate the significance of the data. Following Steel et al. [70], differences between treatments were assessed with a 5% probability using the Tukey HSD test.

## 5. Conclusions

Leaf extract from all the *Moringa oleifera* landraces considerably improved gaseous exchange properties, photosynthetic pigment concentration, enzymatic activities, growth, yield, and quality characteristics under normal and drought conditions. However, MLE derived from Faisalabad landraces showed a higher bio-stimulating potential in improving gaseous exchange properties, increasing photosynthetic pigment concentration, and maximal enzyme activity in rice plants grown under both normal and water scarcity conditions. It can be concluded based on findings of current experimentation that the foliar application of *Moringa oleifera* leaf extract from the Faisalabad landrace can be used to increase the productivity of field crops in normal and water-deficient conditions.

## Figures and Tables

**Figure 1 plants-12-02511-f001:**
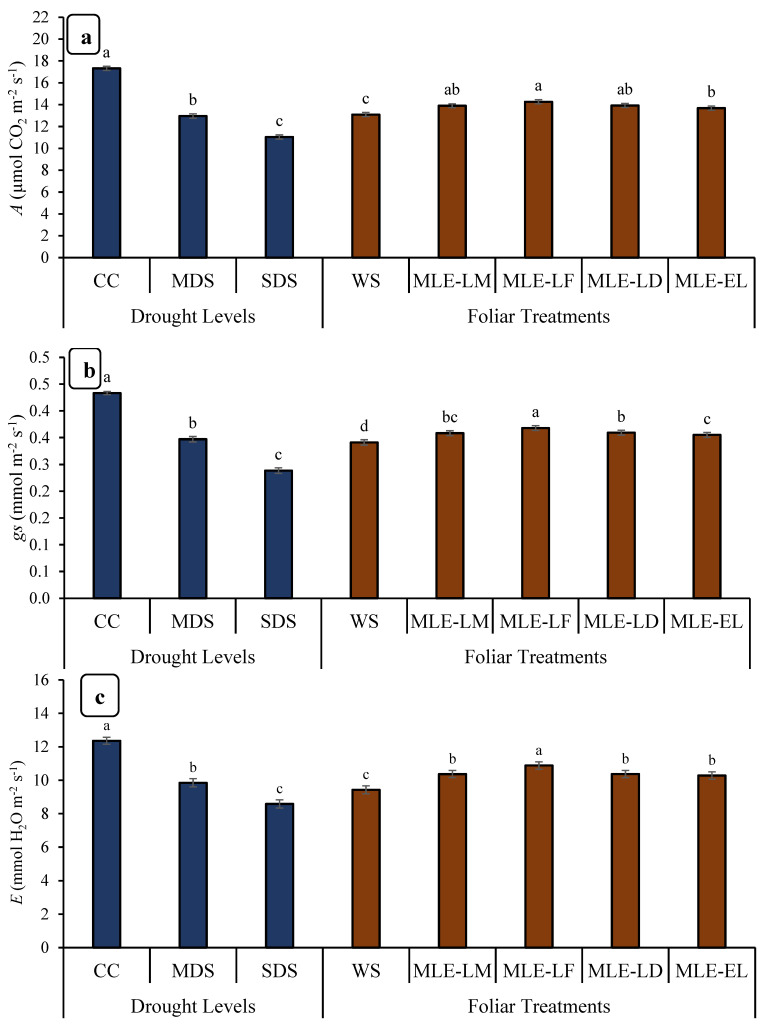
Impact of leaf extracts from local and exotic Moringa landraces on photosynthesis rate (*A*) (**a**), stomatal conductance to water (*gs*) (**b**) and respiration rate (*E*) (**c**) of rice cultivated under normal and water deficit environments (*n* = 4). Bars sharing the same letter did not differ significantly at *p* = 0.05.

**Figure 2 plants-12-02511-f002:**
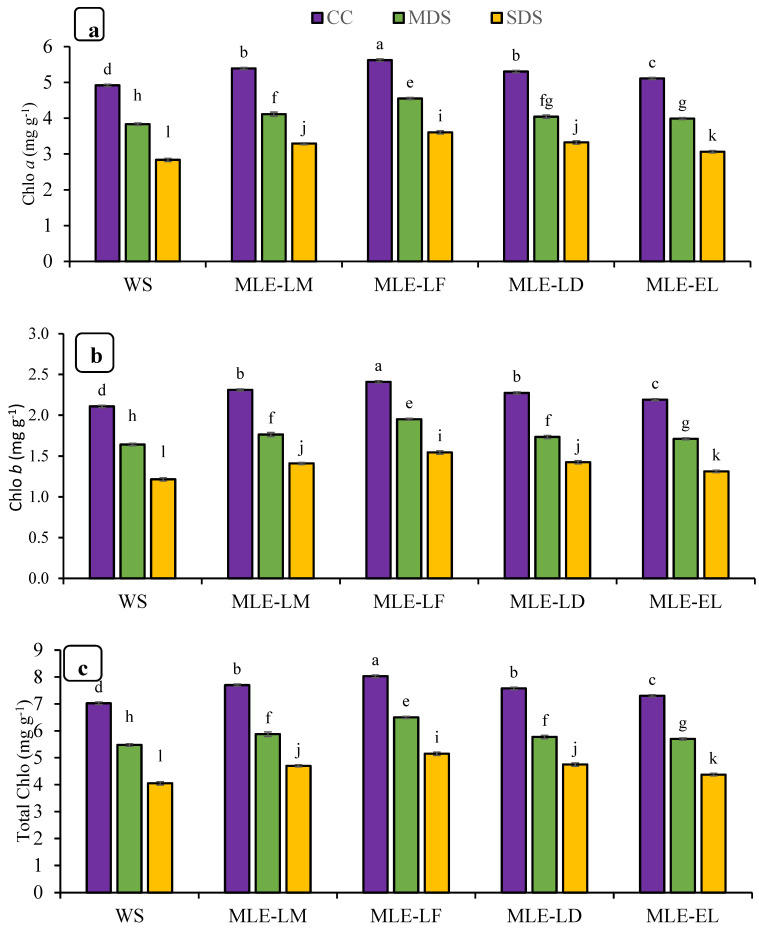
Impact of leaf extracts from local and exotic Moringa landraces on chlorophyll *a* (**a**), chlorophyll *b* (**b**), total chlorophyll (**c**), and carotenoid contents (**d**) of rice leaf cultivated under normal and water deficit environments (*n* = 4). Bars sharing the same letter did not differ significantly at *p* = 0.05.

**Figure 3 plants-12-02511-f003:**
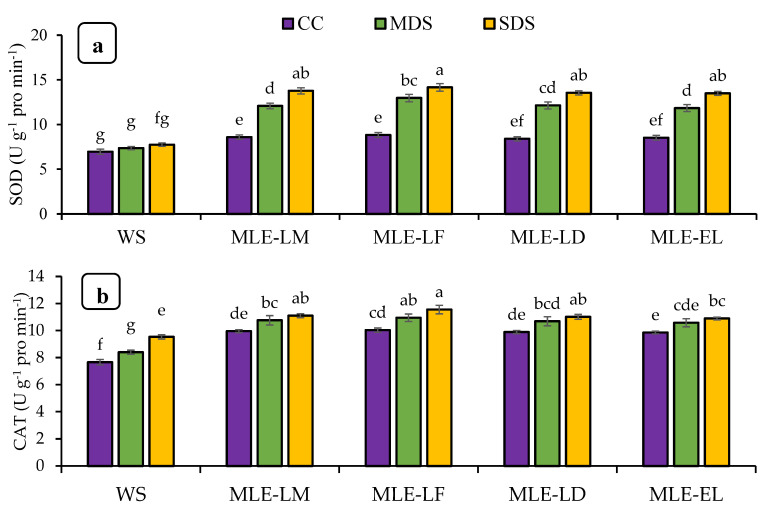
Impact of leaf extracts from local and exotic Moringa landraces on superoxide dismutase (SOD) (**a**), catalase (CAT) (**b**), ascorbate peroxidase (APX) (**c**), and hydrogen peroxide (H2O2) (**d**) of rice leaf cultivated under normal and water deficit environments (*n* = 4). Bars sharing the same letter did not differ significantly at *p* = 0.05.

**Figure 4 plants-12-02511-f004:**
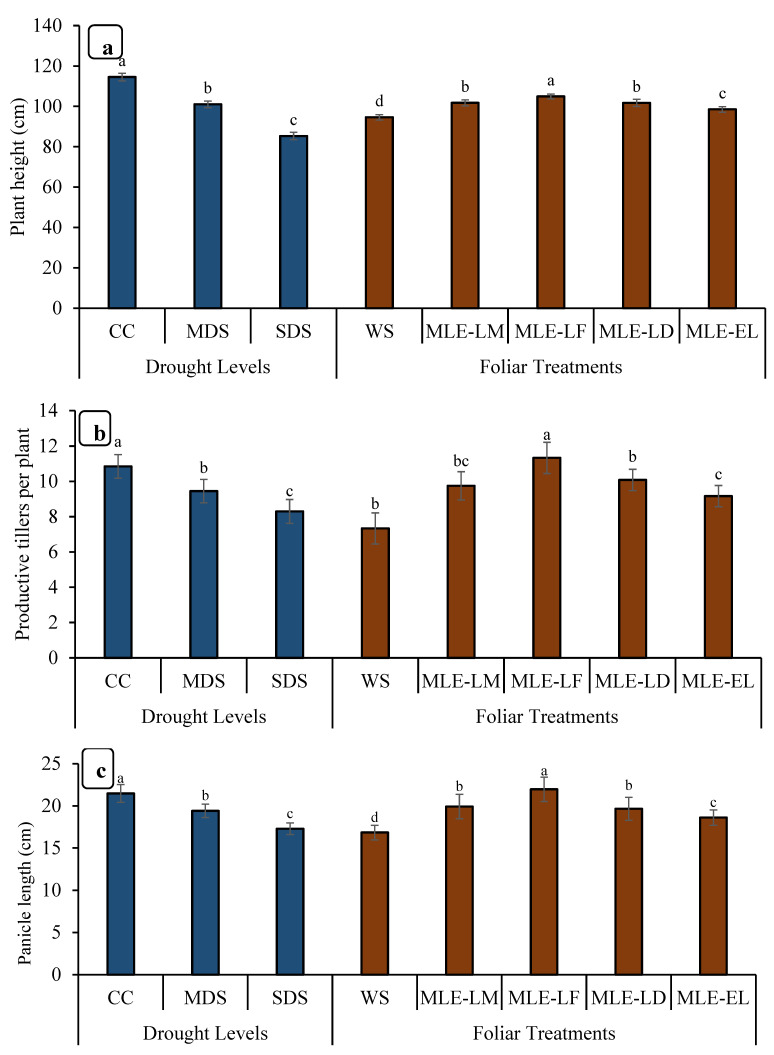
Impact of leaf extracts from local and exotic Moringa landraces on plant height (**a**), number of productive tillers per plant (**b**) and panicle length (**c**) of rice plant cultivated under normal and water deficit environments (*n* = 4). Bars sharing the same letter did not differ significantly at *p* = 0.05.

**Figure 5 plants-12-02511-f005:**
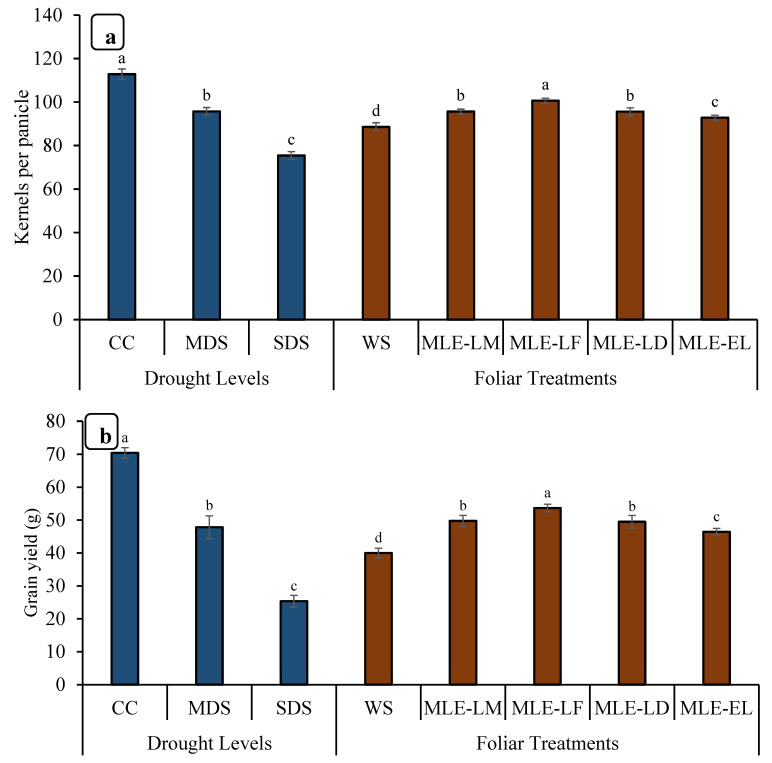
Impact of leaf extracts from local and exotic Moringa landraces on the number of kernels per panicle (**a**) and grain yield per plant (**b**) of rice cultivated under normal and water deficit environments (*n* = 4). Bars sharing the same letter did not differ significantly at *p* = 0.05.

**Figure 6 plants-12-02511-f006:**
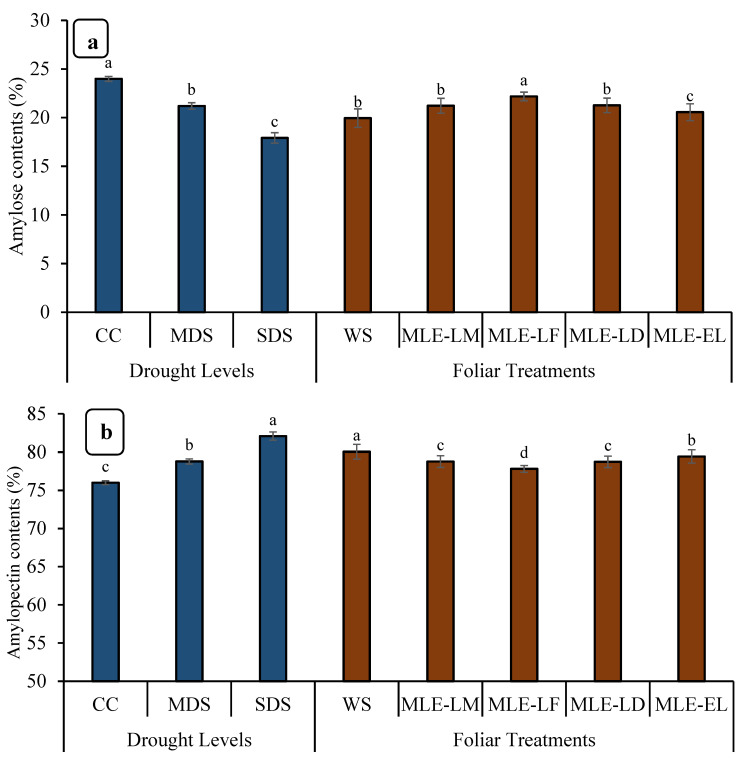
Impact of leaf extracts from local and exotic Moringa landraces on amylose contents (**a**) and amylopectin contents (**b**) of rice grain cultivated under normal and water deficit environments (*n* = 4). Bars sharing the same letter did not differ significantly at *p* = 0.05.

**Table 1 plants-12-02511-t001:** Analysis of variance for gas exchange, physiological, enzymatic, agronomic, yield, and quality attributes of rice plants cultivated under different levels of drought stress and foliar application of Moringa leaf extract (*n* = 4).

SOV	DF	*A* (µmol CO_2_ m^−2^ s^−1^)	*gs* (mmol m^−2^ s^−1^)	*E* (mmol H_2_O m^−2^ s^−1^)	Chlo *a* (mg g^−1^)	Chlo *b* (mg g^−1^)	Total Chlo (mg g^−1^)
WT	2	206.8 **	73.98 *	0.152 *	21.03 *	3.865 *	42.93 *
FT	4	2.264 *	3.294 *	0.002 *	0.881 *	0.162 *	1.801 *
WT × FT	8	0.030 ^NS^	0.018 ^NS^	0.001 ^NS^	0.016 *	0.003 *	0.034 *
SOV	DF	Carotenoids (mg g^−1^)	SOD (U g^−1^ pro min^−1^)	CAT (U g^−1^ pro min^−1^)	APX (U g^−1^ pro min^−1^)	H_2_O_2_ (µmole g^−1^)	Plant Height (cm)
WT	2	1.716 *	83.65 *	1.841 *	1.274 *	499.3 **	4278 *
FT	4	0.072 *	42.64 *	10.58 *	7.272 *	35.32 *	183.9 *
WT × FT	8	0.001 *	7.462 *	1.981 *	1.603 *	6.612 *	1.101 ^NS^
SOV	DF	Kernels per Panicle (Number)	Panicle Length (cm)	Tillers per Plant (number)	Grain Yield (g)	Amylose (%)	Amylopectin (%)
WT	2	7009 *	87.68 *	32.62 *	10,156 *	184.6 *	184.6 **
FT	4	234.6 *	42.27 *	25.69 *	311.8 *	8.415 *	8.415 *
WT × FT	8	4.401 ^NS^	1.074 ^NS^	0.617 ^NS^	25.20 ^NS^	0.479 ^NS^	0.479 ^NS^

SOV = source of variance; WT = water treatments; FT = foliar treatments; WT × FT = interaction of water and foliar treatments; DF = degree of freedom; *A* = photosynthesis rate; *gs* = stomatal conductance to water; *E* = respiration rate; Chlo = chlorophyll; SOD = superoxide dismutase; CAT = catalase; APX = ascorbate peroxidase. * = significant at *p* < 0.05; ** = significant at *p* < 0.01; NS = statistically non-significant.

## Data Availability

The data that support the outcomes of the current experimentation are available from the corresponding authors (S.K. and Z.H.) upon reasonable request.

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
