# Peer review of "Moringa Leaf Extract Mitigates the Adverse Impacts of Drought and Improves the Yield and Grain Quality of Rice through Enhanced Physiological, Biochemical, and Antioxidant Activities"

_plants, 2023, doi:10.3390/plants12132511_

Round 1
Reviewer 1 Report
Please find my comments in the attached pdf file.

Author Response
Response Sheet
Subject: Response to Comments
Title: Moringa Leaf Extract Mitigates the Adverse Impacts of Drought and Improves the Yield and Grain Quality of Rice through Enhanced Physiological, Biochemical and Antioxidant Activities.
We are very much thankful to reviewer for sparing time to review our manuscript and provided valuable comments and suggestions for the improvement of the manuscript.
I, Shahbaz Khan, corresponding author of the manuscript, am enclosing herewith a revised manuscript entitled “Moringa Leaf Extract Mitigates the Adverse Impacts of Drought and Improves the Yield and Grain Quality of Rice through Enhanced Physiological, Biochemical and Antioxidant Activities” for publication in “Plants” after possible improvements. All the comments and suggestions are addressed accordingly and incorporated in the revised manuscript. Details of individual comments are given below.
Comment: There are many previous studies were handling with this topic. What is the novelty of this research?
Response: We say thanks to you for investing your time to review the manuscript. You are right that there are many studies with this topic but it is a first study in which biostimulant potential of leaves extract from various Moringa oleifera landraces are studied regarding rice grown under water deficit conditions.
Comment: Add concentration.
Response: Suggestions are incorporated and highlighted in the revised manuscript.
Comment: Use words different than title.
Response: Suggestions are incorporated and highlighted in the revised manuscript.
Comment: Add reference.
Response: Suggestions are incorporated and highlighted in the revised manuscript.
Comment: Add reference.
Response: Suggestions are incorporated and highlighted in the revised manuscript.
Comment: Add reference.
Response: Suggestions are incorporated and highlighted in the revised manuscript.
Comment: Add reference.
Response: Suggestions are incorporated and highlighted in the revised manuscript.
Comment: If you know that, why you did your work?
Response: Regarding rice crop, it was not investigated earlier.
Comment: Add measuring units for all parameters.
Response: Suggestions are incorporated and highlighted in the revised manuscript.
Comment: Where is the discussion? it is just compare with other works!!
Response: Suggestions are incorporated and highlighted in the revised manuscript.
Comment: Where is the discussion? it is just compare with other works!!
Response: Suggestions are incorporated and highlighted in the revised manuscript.
Comment: Where is the discussion? it is just compare with other works!!
Response: Suggestions are incorporated and highlighted in the revised manuscript.
Comment: you discussed the effect of drought but you don’t discuss the effect of moringa??
Response: Suggestions are incorporated and highlighted in the revised manuscript.
Comment: Italic.
Response: Suggestions are incorporated and highlighted in the revised manuscript.
Comment: Where is the discussion? it is just compare with other works!!
Response: Suggestions are incorporated and highlighted in the revised manuscript.
Comment: Year of experiment?
Response: Suggestions are incorporated and highlighted in the revised manuscript.
Comment: How you measure??
Response: Field capacity was determined by the gravimetric method according to Nachabe (1998). According to the methods, pots were weighed regularly on daily basis and measured amount of water was added accordingly.
Comment: More details are required for these methods.
Response: Suggestions are incorporated and highlighted in the revised manuscript.
A revised manuscript with highlights is attached here for your kind consideration. We say again thanks for your valuable comments and suggestions.

Reviewer 2 Report
Please incorporate all comments.

please improve English quality in all sections.
Author Response
Response Sheet
Subject: Response to Comments
Title: Moringa Leaf Extract Mitigates the Adverse Impacts of Drought and Improves the Yield and Grain Quality of Rice through Enhanced Physiological, Biochemical and Antioxidant Activities.
We are very much thankful to reviewer for sparing time to review our manuscript and provided valuable comments and suggestions for the improvement of the manuscript.
I, Shahbaz Khan, corresponding author of the manuscript, am enclosing herewith a revised manuscript entitled “Moringa Leaf Extract Mitigates the Adverse Impacts of Drought and Improves the Yield and Grain Quality of Rice through Enhanced Physiological, Biochemical and Antioxidant Activities” for publication in “Plants” after possible improvements. All the comments and suggestions are addressed accordingly and incorporated in the revised manuscript. Details of individual comments are given below.
Comment: Line 16: Please rewrite it.
Response: The sentence is rewritten and incorporated in the revised manuscript highlighted with track changes.
Comment: Line 20: “Present was designed” what is this. Please correct it.
Response: Suggestions are incorporated and highlighted in the revised manuscript.
Comment: Line: 27: “under normal and unfavorable growth conditions” what do you mean? How can you prove that MLE increased the growth in these conditions? Please explain.
Response: Normal growth conditions mean control condition in which there was no drought stress while in unfavorable growth conditions, drought stress was imposed as mild and severe. Further details are given in materials and methods section 4.2; There were three levels of drought i.e. control conditions (CC), moderate drought stress (MDS) and severe drought stress (SDS). Drought treatments; CC, MDS and SDS, were induced by limiting the water supply to keep the field capacity of pots at 100%, 75% and 50% respectively. Field capacity was determined by the gravimetric method according to Nachabe (1998).
Regarding second part of the comment, how can you prove that MLE increased the growth in these conditions? This statement is given on the basis of outcomes of study. You can observe the results please.
Comment: Line 27: why only one MLE land traces is described. Please provide results of all land races.
Response: All the MLE landraces improved the growth but maximum potential was observed in the leaf extract from Faisalabad landrace. To maintain the range of number of words in abstract, main findings are presented in the abstract.
Comment: Line 28: Please write increase percentages of all your findings.
Response: Suggestions are incorporated and highlighted in the revised manuscript.
Comment: Line 31: Please write conclusion again.
Response: Conclusion is rewritten and improved accordingly. Suggestions are incorporated and highlighted in the revised manuscript.
Comment: Line 38: Please rewrite it into meaningful.
Response: Sentence is rewritten. Suggestions are incorporated and highlighted in the revised manuscript.
Comment: Line 41: “Drought stress is the main factor limiting agricultural production in the rain-41 fed regions of Asia, which covers more than 23 million hectares” what authors want to describe here please clarify and 23 million hectares what? Please provide latest reference here.
Response: Suggestions are incorporated and highlighted in the revised manuscript.
Comment: Line 48: Worldwide, rice grain yield was recorded 514.8 million 48 tons from and area of 165.25 million hectares? Please verify it annually or?
Response: It is verified and updated. Suggestions are incorporated and highlighted in the revised manuscript.
Comment: Line 51-53: However, for 51 ROS detoxification, various enzymatic antioxidants such as catalase (CAT), super dis-52 mutase (SOD) and peroxidase (POD) along with non-enzymatic antioxidants plays an Important role. What type of role author want to describe please mention.
Response: Suggestions are incorporated and highlighted in the revised manuscript.
Comment: Line 53: antioxidants plays an important role. Please correct it grammatically.
Response: Suggestions are incorporated and highlighted in the revised manuscript.
Comment: Line 56: According to Nawaz et al. (2013), drought stress also reduces the rate and time 56 of grain filling which is ultimately responsible for reduction in grain yield. Please replace it with latest one.
Response: Suggestions are incorporated and highlighted in the revised manuscript.
Comment: Line 79-85: Please write latest literature study about use of MLE to mitigate the drought stress in different plants.
Response: Suggestions are incorporated and highlighted in the revised manuscript.
Comment: Figure 1 a, b, Figure 2 b and c, Figure 3 b and c, Figure 4 a and b, Figure 5 a, and Figure 6 a are not appropriate showing clear concept. Please change them.
Response: We have changed axis limits (Figure 1 a and b) to make it clearer. The alphabets at the top of each bar depict the statistical difference among the treatments. The bars having the same alphabet are statistically similar while bars having different alphabets are statistically significant. In figure 1, 4, 5 and 6, first (left side) three bar represent the drought levels and next five bars (on the right) show the foliar treatments. Suggestions are incorporated and highlighted in the revised manuscript. In case of figure 2 and 3, under a foliar treatment, impact of three drought levels is depicted with three bars as interaction of drought and foliar treatments was observed statistically significant (Table 1) regarding the parameters presented in these figures.
Comment: How did you apply MLEs on which rice plants? Normal, drought or both conditions, please explain.
Response: MLE treatments were applied on plants under normal and drought conditions but there was a control experimental unit in normal and drought conditions where MLE was not applied.
Comment: Line 201-202: Results obtained of the current experiment showed 201 that A, gs and E were significantly reduced under water deficit conditions (Fig. 1a–c). Modify it
Response: Suggestions are incorporated and highlighted in the revised manuscript.
Comment: Line 207-220: During discussion please provide the evidence of increase chlorophyll contents using MLEs on other crops. Also mention about effect of different treatments on rice under drought stress regarding chlorophyll contents.
Response: Suggestions are incorporated and highlighted in the revised manuscript.
Comment: Experimental design is poor not mention clearly. Formulate in table form.
Response: Experimental design is interpreted with details in the Materials and Methods section; 4.9. Statistical Analysis. Suggestions are incorporated and highlighted in the revised manuscript.
Comment: Line 317: Mature, fresh, healthy and disease free leaves from the trees of respective moringa landraces and rinsed with tap water to move the dust. Please mention the specie of moringa.
Response: Suggestions are incorporated and highlighted in the revised manuscript (4.3. Extract Preparation and Application).
Comment: Line 318: After storage of overnight in refrigerator “At which temperature please provide reference.
Response: Leaves are normally placed at 0℃ in the refrigerator just to ice crystal formation. These ice crystals/needles rapture the cell wall to ooze out the sap from the leaves.
Comment: Line 322: Please mention GPS location of all land races.
Response: Suggestions are incorporated and highlighted in the revised manuscript.
Comment: Line 324: exotic landrace of India (MLE-EL). How did author obtain.
Response: Exotic landrace; PKM-1 is well-established at research farm area of Agronomy Department, University of Agriculture, Faisalabad, Pakistan. Suggestions are incorporated and highlighted in the revised manuscript.
Comment: Line 321: How did you obtain 3% concentration of MLE?
Response: A locally assembled machine was used for extraction purpose. The juice/extract obtained directly from the machines was considered as 100%. 3 ml of 100% concentrated extract was added in 97 ml of water to make ml 100 ml of 3% MLE solution.
Comment: Line 324-325: The MLEs in all the treatments were applied thrice during the course of experimentation at tillering, panicle initation and grain filling stages of rice plants. Please provide literature.
Response: There are critical stages in the life cycle of each crop. The tillering, panicle initiation and grain filling stages are considered as critical stages in rice. So, foliar treatments were applied at critical stages.
Comment: Please rewrite Conclusions
Response: Suggestions are incorporated and highlighted in the revised manuscript.
A revised manuscript with highlights is attached here for your kind consideration. We say again thanks for your valuable comments and suggestions.

Round 2
Reviewer 1 Report
Thank you for your improvement
Author Response
Dear Editor,
Thank you so much for sparing the time to review the manuscript.
Thanks again.
Reviewer 2 Report
Thank you very much for your addressing comments but find some comments that are very necessary and can be improved. Please address all comments it will increase consistency and quality of paper.
Thanks.

Please improve your English quality.
Author Response
Response Sheet
Subject: Response to Comments
Title: Moringa Leaf Extract Mitigates the Adverse Impacts of Drought and Improves the Yield and Grain Quality of Rice through Enhanced Physiological, Biochemical and Antioxidant Activities.
We are very much thankful to reviewer for sparing time to review our manuscript and provided valuable comments and suggestions for the improvement of the manuscript.
I, Shahbaz Khan, corresponding author of the manuscript, am enclosing herewith a revised manuscript entitled “Moringa Leaf Extract Mitigates the Adverse Impacts of Drought and Improves the Yield and Grain Quality of Rice through Enhanced Physiological, Biochemical and Antioxidant Activities” for publication in “Plants” after possible improvements. All the comments and suggestions are addressed accordingly and incorporated in the revised manuscript. Details of individual comments are given below.
Comment: “All the MLE landraces improved the growth but maximum potential was observed in the leaf extract from Faisalabad landrace. To maintain the range of number of words in abstract, main findings are presented in the abstract”. I am disagree with this statement. Please highest and lowest outcomes of MLE landraces must add in the abstract.
Response: Suggestions are incorporated and highlighted in the revised manuscript.
Comment: I am not satisfied with graphs without x –axis titles. Please must add.
Response: x–axis titles are added. Suggestions are incorporated and highlighted in the revised manuscript.
Comment: Please write Figure a, b, c etc. either on right or left side of graphs for consistency and quality of paper. You can see in Figure 1, 2, 4, 5 lettering is on left side and in others figures it is on right side. Please make consistency.
Response: Lettering is done on left side in all figures. Suggestions are incorporated and highlighted in the revised manuscript.
Comment: The current experiment was conduction according to completely randomized design (CRD) with factorial arrangement because there were two factors; a-drought levers and b MLE foliar treatments. There were three levels of drought stress i.e. control conditions (CC), moderate drought stress (MDS) and severe drought stress (SDS). Moringa leaf extract (MLE) application consisted of five treatments, i) water spray (WS), ii) MLE from Multan landrace (MLE-LM), iii) MLE from Faisalabad landrace (MLE-LF), iv) MLE from Dera Ghazi Khan Landrace (MLE-LD) and v) MLE from exotic landrace of India (MLE EL). Please write it after 4.1 section.
Response: Suggestions are incorporated and highlighted in the revised manuscript.
Comment: “Leaves are normally placed at 0℃ in the refrigerator just to ice crystal formation. These ice crystals/needles rapture the cell wall to ooze out the sap from the leaves” How did you adjust temperature?
Response: It was an adjustable refrigerator up to -20℃. So, we adjusted it at 0℃.
Comment: Line 280-281: Our findings suggest that the increase in SOD, CAT and APX. Please don’t write antioxidant enzymes names in abbreviation.
Response: Suggestions are incorporated and highlighted in the revised manuscript.
Comment: Please provide photos of experimental work.
Response: Photos of experimental work are not available. In future studies, your suggestion will be considered.
Regarding the English Language improvement, we got help from the researcher of Colorado State University. He reviewed it very critically and draft was improved accordingly. Furthermore, English is also improved by the MDPI experts.
A revised manuscript with highlights is attached here for your kind consideration. We say again thanks for your valuable comments and suggestions.
